👁 | **Open Peer Review** | Environmental Microbiology | Observation

# Conversion of methane to organic acids is a widely found trait among gammaproteobacterial methanotrophs of freshwater lake and pond ecosystems

Ramita Khanongnuch,[1] Rahul Mangayil,[1,2] Antti Juhani Rissanen[1,3]

**ABSTRACT** Aerobic gammaproteobacterial methanotrophs (gMOB) are key organisms controlling methane fluxes at the oxic-anoxic interfaces of freshwater ecosystems. Under hypoxic environments, gMOB may shift their aerobic metabolism to fermentation, resulting in the production of extracellular organic acids. We recently isolated a gMOB strain representing the *Methylobacter* spp. of boreal lake water columns (i.e., *Methylobacter* sp. S3L5C) and demonstrated that it converts methane to organic acids (acetate, formate, malate, and propionate) under hypoxic conditions. Annotation for putative genes encoding organic acid production within the isolate's genome and in environmental metagenome-assembled genomes (MAGs) representing *Methylobacter* spp. suggests that the potential for methane conversion into organic acids is widely found among *Methylobacter* spp. of freshwater ecosystems. However, it is not known yet whether the capability to convert methane to organic acids is restricted to *Methylobacter* spp. or ubiquitously present among other freshwater gMOB genera. Therefore, we isolated representatives of two additional gMOB genera from the boreal lake water columns, i.e., *Methylomonas paludis* S2AM and *Methylovulum psychrotolerans* S1L, and demonstrated similar bioconversion capacities. These genera could convert methane to organic acids, including acetate, formate, succinate, and malate. Additionally, S2AM produced lactate. Furthermore, we detected genes encoding organic acid production within their genomes and in MAGs representing *Methylomonas* spp. and *Methylovulum* spp. of lake and pond ecosystems. Altogether, our results demonstrate that methane conversion to various organic acids is a widely found trait among lake and pond gMOB, highlighting their role as pivotal mediators of methane carbon into microbial food webs of freshwater lake and pond ecosystems.

**IMPORTANCE** Aerobic gammaproteobacterial methanotrophic bacteria (gMOB) play an important role in reducing methane emissions from freshwater ecosystems. In hypoxic conditions prevalent near oxic-anoxic interfaces, gMOB potentially shift their metabolism to fermentation, resulting in the conversion of methane to extracellular organic acids, which would serve as substrates for non-methanotrophic microbes. We intended to assess the prevalence of fermentation traits among freshwater gMOB. Therefore, we isolated two strains representing relevant freshwater gMOB genera, i.e., *Methylovulum* and *Methylomonas*, from boreal lakes, experimentally showed that they convert methane to organic acids and demonstrated via metagenomics that the fermentation potential is widely dispersed among lake and pond representatives of these genera. Combined with our recent study showing coherent results from another relevant freshwater gMOB genus, i.e., *Methylobacter*, we conclude that the conversion of methane to organic acids is a widely found trait among freshwater gMOB, highlighting their role as pivotal mediators of methane carbon into microbial food webs.

Address correspondence to Ramita Khanongnuch, ramita.khanongnuch@tuni.fi, or Antti Juhani Rissanen, antti.rissanen@tuni.fi.

The authors declare no conflict of interest.

See the funding table on p. 5.

10.1128/spectrum.01742-23 **1**

**KEYWORDS** greenhouse gas, climate change, food web, extracellular metabolites, bioconversion, psychrophilic, psychrotolerant

Gammaproteobacterial methanotrophs (gMOB) of several genera, e.g., *Methylobacter*, *Crenothrix*, *Methylomonas*, and *Methylovulum*, are key organisms controlling methane ($CH_4$) fluxes at the oxic-anoxic interfaces of freshwater lake and pond ecosystems, where they can constitute over 50% of prokaryotes (1–3). Being obligate aerobic in nature, gMOB generally require $O_2$ as the electron acceptor to oxidize $CH_4$ into biomass and $CO_2$. At the oxic-anoxic interface, however, they face fluctuating oxygen conditions and occasional hypoxia (i.e., oxygen limitation). During hypoxic conditions, gMOB may shift their cellular metabolism toward fermentation by generating various extracellular organic acids, as shown with a haloalkalitolerant strain *Methylotuvimicrobium alcaliphilum* 20Z (4). The $CH_4$-derived organic acids could then serve as growth substrates for heterotrophic and methylotrophic bacteria in microbial food webs (5–7). Besides methanotroph biomass carbon, an important component of food webs until the top consumer level (8, 9), other microbes consuming $CH_4$-derived soluble compounds potentially play a role in channeling $CH_4$-carbon to consumers. We have recently shown that the potential for organic acid production is also found among freshwater lake gMOB (10). We isolated a psychrophilic gMOB strain representing genus *Methylobacter* (i.e., *Methylobacter* sp. S3L5C) from the water column of a boreal lake, demonstrated the bacterium's capacity for the bioconversion of $CH_4$ to organic acids, and predicted the putative genes (enzymes) driving this process (10). Furthermore, based on the analyses of metagenome-assembled genomes (MAGs), we concluded that the genetic potential to produce organic acids is a widely found trait among *Methylobacter* spp. in freshwater ecosystems (10–12), indicating their role as critical mediators regulating the bioconversion of $CH_4$ to organic acids in freshwater ecosystems (5–7). However, to date, similar observations for other freshwater lake gMOB genera have not yet been reported. We hereby aim to demonstrate that the capability to convert $CH_4$ to organic acids is not restricted to *Methylobacter* spp. but exists among other gMOB genera in freshwater lake and pond ecosystems.

To address our aim, we isolated representatives of two additional freshwater lake gMOB genera, i.e., *Methylovulum psychrotolerans* S1L and *Methylomonas paludis* S2AM, from hypoxic water column layers of $O_2$-stratified boreal lakes located in Southern Finland (Table 1, pictures on the colonies and cells in Fig. S1). The strains' isolation, genome sequencing, and phylogenetic assignment were described previously (13). The genes in their genomes and representative MAGs of metagenomic operational taxonomic units representing *Methylomonas* spp. and *Methylovulum* spp. of boreal and subarctic lakes and ponds as well as one temperate lake and one tropical reservoir [MAGs assembled and taxonomically annotated by Buck et al. (14)] were predicted using Prodigal (v. 2.6.3) (15) and annotated according to Kyoto Encyclopedia of Genes and Genomes using KofamKOALA (https://www.genome.jp/tools/kofamkoala/; accessed 27 February 2023) (16). We specifically focused on the key genes encoding enzymes involved in organic acid and $H_2$ production. The optimum growth conditions of the strains were determined in batch tests at different pH, temperatures, and nitrogen sources (see detailed methods in Supplementary Information) (Table 1; Fig. S2 and S3). In addition, the isolates' capacity to generate organic acids was demonstrated in specific batch tests (six bottles per strain) as described in Khanongnuch et al. (10). Briefly, S1L and S2AM were grown in nitrate mineral salt medium and incubated at 23°C. For three bottles in the experimental setup, the initial headspace [containing 20% $CH_4$ + 80% air (vol/vol)] was replenished with the original headspace content at days 10 and 14 for S1L and S2AM, respectively. As a control, the remaining experimental bottles were left without headspace replenishment, and the incubation was continued until days 20 and 34 for S1L and S2AM, respectively. During incubation, the cell growth, gaseous content, and organic acids were periodically monitored (see detailed methods in Supplementary Information) (Fig. 1).

**TABLE 1** Characteristics of the gMOB isolates

| Strain | *Methylovulum psychrotolerans* S1L | *Methylomonas paludis* S2AM | *Methylobacter* sp. S3L5C |
|---|---|---|---|
| Cell morphology | Cocci | Rods | Cocci |
| Cell size (µm) | 1.0–1.8 diameter | 0.7–1.2 × 1.4–3.0 | 1.7–4.0 diameter |
| Optimal temperature (growth) (°C)[a] | 20–24 (4–30) | 15–27 (0.2–30) | 8–12(0.1–20) |
| Optimal pH (growth) | 7.4 (4.7–8.3) | 6.0–6.9 (5.0–7.5) | 6.0–7.3 (6.0–8.3) |
| *nif* Gene | Yes | Yes | Yes |
| Motility[b] | – | – | – |
| Pigmentation | Pale pink | Pale pink | – |
| Excreted organic acid compounds | Acetate, formate, malate, and succinate | Acetate, formate, malate, succinate, and lactate | Acetate, formate, malate, and propionate |
| Carbon conversion efficiency of consumed methane into total accumulated organic acids[c] | | | |
| Acetate-C | 0.7 | 2.7 | 2.4 |
| Formate-C | 0.1 | 0.4 | <0.1 |
| Malate-C | 0.1 | 0.1 | <0.1 |
| Succinate-C | <0.1 | 0.1 | – |
| Lactate-C | – | 0.3 | – |
| Propionate-C | – | – | 0.1 |
| | 0.9% | 3.6% | 2.5% |
| Source | Lake water layer (Lovojärvi, Finland) | Lake water layer (Alinen Mustajärvi, Finland) | Lake water layer (Lovojärvi, Finland) |
| Reference | This study | This study | Khanongnuch et al. 2022 (10) |

[a]Based on the temperature test, S1L and S2AM are psychrotolerant, while S3L5C is psychrophilic.
[b]–, not detected.
[c]Organic acid accumulation at the end of the test with CH4 and air replenishment. See the calculation in supplemental data for Fig. 1 in the sections C,E-S1L-GC and D,F-S2AM-GC.

Strains S1L and S2AM were psychrotolerant and enabled to use of nitrate and ammonium as nitrogen sources (Table 1; Fig. S2 and S3). Both strains produced acetate, formate, malate, and succinate, while S2AM also produced lactate (up to 0.2 µM) in the subsequent specific tests to demonstrate their organic acid production (Table 1; Fig. 1G through J). Similar as noticed for S3L5C as reported by Khanongnuch et al. (10), acetate was the most prominent metabolite, up to 0.8 µM and 2.9 µM, for S1L and S2AM, respectively, in these specific batch tests. It was followed by formate, up to 0.1 µM and 0.9 µM, for S1L and S2AM, respectively, while the other products had lower concentrations, < 0.1 µM (Fig. 1G–J). The average consumed $O_2/CH_4$ ratio (~1.0) was below the stoichiometric ratio in aerobic $CH_4$ oxidation (Fig. 1E and F). This indicates that $O_2$-limited $CH_4$ oxidation (during hypoxic conditions) initiated the accumulation of organic acids (4, 10). For S1L, the growth and accumulation of organic acids were generally higher in the treatment with headspace gases replenished (Fig. 1A, G and I; Fig. S4A) (growth: $P < 0.01$, organic acids: $P < 0.01$, see Supplementary data for Fig. 1 and Fig. S4), agreeing with results from our previous study of *Methylobacter* sp. S3L5C (10). For S2AM, the gas replenishment did not improve growth or organic acid accumulation (Fig. 1B, H and J; Fig. S4B) (growth: $P = 0.57$, organic acids: $P = 0.51$, see Supplementary data for Fig. 1; Fig. S4), likely due to the high viscosity visually observed in the liquid medium, causing mass transfer limitation on $CH_4$ uptake.

The genes encoding putative enzymes driving the organic acid production were found in the genomes of both strains (Table S1; Fig. S5). As further proof of functions under fermentative conditions, both strains contained genes encoding $H_2$-producing enzymes (Table S1), as did S3L5C (10). Surprisingly, lactate was observed during incubation of S2AM (Fig. 1H and J); however, its genome did not encode an identifiable lactate dehydrogenase. It is possible that lactate excretion is from methylglyoxal/2-oxopropanal detoxification generally occurring in microorganisms (17, 18). This detoxification to D-lactate was potentially carried out by the products of the *gloA* and *gloB* genes found in S2AM (Table S1), responding with the observation in other methanotrophs

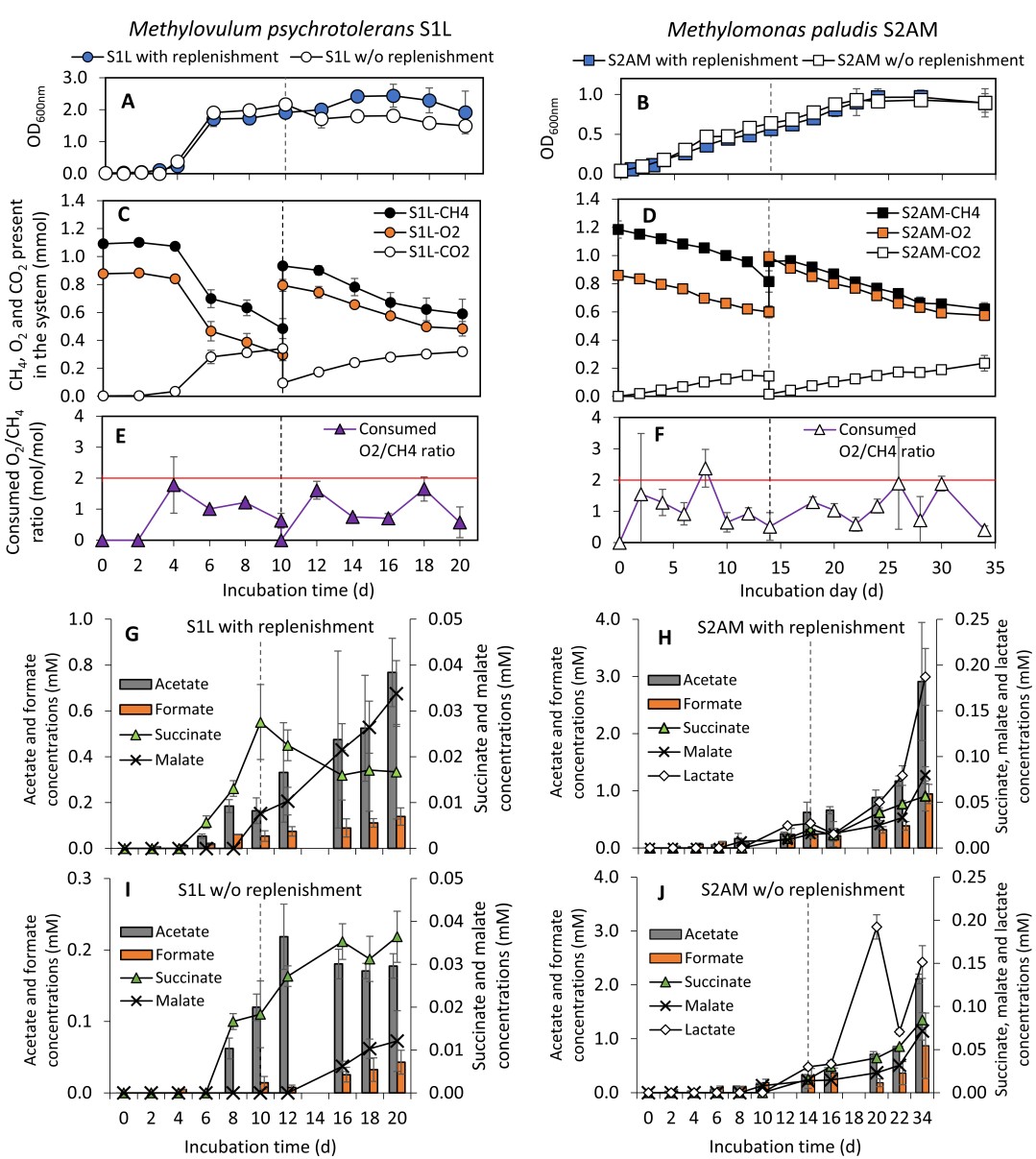

**FIG 1** Performance on $CH_4$ oxidation and organic acid excretion of *Methylovulum psychrotolerans* S1L (left) and *Methylomonas paludis* S2AM (right) during the test with and without $CH_4$ and air replenishment on days 10 and 14 (vertical dot line) for strains S1L and S2AM, respectively. The profiles of (A, B) cell growth during the test with and without the gas replenishment, as well as the profiles of (C, D) $CH_4$ and $O_2$ utilization and $CO_2$ production, and (E, F) consumed $O_2/CH_4$ molar ratio during the test with the gas replenishment. The red horizontal line (E, F) indicates the stoichiometric $O_2/CH_4$ ratio for aerobic methane oxidation ($CH_4 + 2O_2 \rightarrow CO_2 + 2H_2O$). (G, H) Organic acid excretion profile during the test with the replenishment and (I, J) without the replenishment. The error bars represent the standard deviation among the biological triplicate samples.

(19, 20). However, this observation requires further experimental validations, and the methylglyoxal formation in methanotrophs has not been elucidated (20). Our MAG analyses also indicate that the genetic potential of *Methylomonas* spp. and *Methylovulum* spp. for organic acid and $H_2$ production is widely dispersed in boreal and subarctic lakes and ponds (Finland, Sweden, and Canada) and also found within the temperate lake (Switzerland) and tropical reservoir (Puerto Rico) in the MAG data set (Table S1), similar as noticed for *Methylobacter* spp. in environmental samples (10–12).

Altogether, our experiments with gMOB strains representing three genera (Table 1) and MAG analyses demonstrate that the ability to convert $CH_4$ to various organic acids is a prevalent trait among lake and pond gMOB. Hence, gMOB are important mediators

in incorporating $CH_4$-carbon into microbial food webs of freshwater lake and pond ecosystems.

## ACKNOWLEDGMENTS

The authors thank Prof. Mette Marianne Svenning and Anne Grethe Hestnes, The Arctic University of Norway, Tromsø, Norway, for their guidance and support in methanotroph isolation and cultivation. The authors also thank the staff at Lammi Biological Station (Finland) for their support in sampling. Reviewers are acknowledged for their valuable comments and suggestions that improved the paper.

This study was funded by the Academy of Finland (Grant no. 346751 for A.J.R., 353750 for A.J.R. and R.K., and 346983 for R.M.) and Kone Foundation (Grant no. 201803224 for A.J.R. and R.K.). Open access funding was provided by Tampere University.

R.K.: Conceptualization, Methodology, Formal analysis, Investigation, Writing – Original Draft, Writing – Review & Editing, Visualization; R.M.: Conceptualization, Methodology, Writing – Review & Editing, Supervision; A.J.R.: Conceptualization, Formal analysis, Investigation, Resources, Writing – Original Draft, Writing – Review & Editing, Supervision, Project Administration, Funding Acquisition.

## AUTHOR AFFILIATIONS

[1]Faculty of Engineering and Natural Sciences, Tampere University, Tampere, Finland
[2]Department of Bioproducts and Biosystems, School of Chemical Engineering, Aalto University, Espoo, Finland
[3]Natural Resources Institute Finland, Helsinki, Finland

## AUTHOR ORCIDs

Ramita Khanongnuch  http://orcid.org/0000-0001-7679-5928
Antti Juhani Rissanen  http://orcid.org/0000-0002-5678-3361

## FUNDING

| Funder | Grant(s) | Author(s) |
| --- | --- | --- |
| Academy of Finland (AKA) | 346751 | Antti Juhani Rissanen |
| Academy of Finland (AKA) | 353750 | Ramita Khanongnuch |
|  |  | Antti Juhani Rissanen |
| Academy of Finland (AKA) | 346983 | Rahul Mangayil |
| Koneen Säätiö (Kone Foundation) | 201803224 | Ramita Khanongnuch |
|  |  | Antti Juhani Rissanen |

## DATA AVAILABILITY

The research data are available in the supplemental data sets (see Supplemental Material).

## ADDITIONAL FILES

The following material is available online.

Supplemental Material

**Supplemental data for Fig. 1 (Spectrum01742-23-s0001.xlsx).** Supporting data related to Fig. 1.
**Supplemental data for Fig. S2 (Spectrum01742-23-s0002.xlsx).** Supporting data related to Fig. S2.

**Supplemental data for Fig. S3 (Spectrum01742-23-s0003.xlsx).** Supporting data related to Fig. S3.

**Supplemental data for Fig. S4 (Spectrum01742-23-s0004.xlsx).** Supporting data related to Fig. S4.

**Supplemental information (Spectrum01742-23-s0005.docx).** Supplemental methods and Fig. S1 to S5.

**Table S1 (Spectrum01742-23-s0006.xlsx).** Key genes encoding enzymes involved in organic acid and $H_2$ production in *Methylomonas paludis* S2AM and *Methylovulum psychrotolerans* S1L, as well as representative MAGs of mOTU affiliated with *Methylomonas* and *Methylovulum*.

## Open Peer Review

**PEER REVIEW HISTORY (review-history.pdf).** An accounting of the reviewer comments and feedback.

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
