## [Reviewer comments · Microbiology Spectrum]

Microbiology Spectrum

Conversion of methane to organic acids is a widely found trait among gammaproteobacterial methanotrophs of freshwater lake and pond ecosystems

Ramita Khanongnuch, Rahul Mangayil, and Antti Rissanen

Corresponding Author(s): Antti Rissanen, Tampereen yliopisto - Hervannan kampus

Review Timeline:

Submission Date:	April 26, 2023
Editorial Decision:	July 12, 2023
Revision Received:	September 6, 2023
Accepted:	September 9, 2023

Editor: Jannell Bazurto

Reviewer(s): The reviewers have opted to remain anonymous.

Transaction Report:

DOI: <https://doi.org/10.1128/spectrum.01742-23>

July 12, 2023

Dr. Antti Juhani Rissanen
Tampereen yliopisto - Hervannan kampus
Faculty of Engineering and Natural Sciences
Korkeakoulunkatu 6
Tampere FI-33720
Finland

Re: Spectrum01742-23 (The potential to convert methane to organic acids is a widely found trait among gammaproteobacterial methanotrophs of freshwater lake and pond ecosystems)

Dear Dr. Antti Juhani Rissanen:

Many thanks for submitting your manuscript, entitled "The potential to convert methane to organic acids is a widely found trait among gammaproteobacterial methanotrophs of freshwater lake and pond ecosystems" for consideration by Microbiology Spectrum. The expertise from two external referees for the assessment of the manuscript was requested and as you can see from their comments, they agreed that the topic of the manuscript is relevant and the experimentation and data provided are valuable. In general, we do believe that tackling all of the Referees' criticisms through manuscript modification is necessary, please respond fully to each comment by modifying the submission.

Link Not Available

Sincerely,

Jannell Bazurto

Journals Department
Reviewer comments:

Reviewer #1 (Comments for the Author):

This manuscript describes a useful and generally well presented study that shows organic acid production in two isolates of methanotrophs representing genera in which this activity had not previously been seen. There is also useful discussion of

genomic and metagenomic information relating to this activity in the isolates and beyond. A number of issues require attention as indicated below.

1. Line 105. Since two strains are referred to here, the apostrophe should come after the final s at the end of this line.
2. Lines 134 to 139 and Figure 1. I could not find evidence in the figure to support the statements here about the different effect of replenishing the methane/air gas mixture on the two strains characterised in this work. The figure shows only one set of data for optical density for each strain, so the effect of omitting the gas replenishment cannot be judged. Also, the simple statement in the text that replenishment increases organic acid production in strain S1L but not in strain S2AM does not adequately summarise what the data show. In fact, in strain S1L gas replenishment results in more succinate but less malate compared to what happens without replenishment. In strain S2AM it is true that replenishment generally does not increase organic acid production, but there is a sharp spike in lactate concentration at 20 h when no replenishment is done. The OD data with and without has replenishment need to be shown and more accurate description of the data is needed. If the lactate spike in strain S2AM is an artifact (rather than a reproducible phenomenon), some additional experimental work would be needed. Statistical tests to validate any important changes commented upon would be helpful.
3. Legend of Figure 1. The red line referred to in parts E and F is horizontal, not vertical.
4. Line 145. To cover the possibility that lactate dehydrogenases might exist that cannot currently be identified by bioinformatics, I suggest inserting "an identifiable" before "lactate".
5. Line 148. Insert "the products of the" between "by" and "gloA".
6. It would be helpful to include, maybe in the supplementary material, a figure showing the pathways from methane to organic acids together annotated to show which genes have been found in the genomes of strains S1L and S2AM.
7. Mindful that methanotrophs have been considered for conversion of methane into multicarbon acids and other molecules, it would be interesting to calculate the maximum carbon atom percentage of methane to organic acid achieved in this work.
8. Table S1. The information provided about the gene functions from the isolated strains is very clear, but that from the MAGs are given as KO numbers without gene names or putative functions. The table would be more easily accessible if some information about gene name or putative function were added for these.

Reviewer #2 (Comments for the Author):

In this work, the authors investigate the fermentative abilities of two isolated strains of methanotrophic gammaproteobacteria, *Methylomonas paludis* S2AM and *Methylovulum psychrotolerans* S1L. This work follows up on a recent publication from the same authors (10.1038/s43705-022-00172-x) that reports on the fermentative ability of a *Methylobacter* isolate from the same boreal lake environment. These studies are important because they provide molecular information about the ability of methanotrophs to support non-methanotrophic organisms in the environment via methane gas. The work uses the largely the same methods as the previous publication, and the conclusions appear sound and are not overstated.

Comments:

Figure 1:

- I think it would be helpful to put the organic acid legends in panels I and J as well.
- A description of the error bars is not provided. Are these means and standard deviations? How many replicates? This is also true in figures S2 and S3.
- In E and F, the Oxygen:Methane ratio appears to dip below 1 in some cases, which is unusual for aerobic methanotrophic metabolism. The authors should provide an explanation for this.

Methods:

- It would be nice to have more information about how the organic acid quantification was performed (mobile phase, gradient, concentrations of standards, etc.) since this is the main point of the paper.
- I suggest putting the description from the top of table S1 in the methods section for how genomes and MAGs were searched for fermentative genes.

Staff Comments:

Preparing Revision Guidelines

Please return the manuscript within 60 days; if you cannot complete the modification within this time period, please contact me. If you do not wish to modify the manuscript and prefer to submit it to another journal, please notify me of your decision immediately so that the manuscript may be formally withdrawn from consideration by Microbiology Spectrum.

Response to Reviewers

We thank the reviewers for the constructive feedback and recognition of our study's contributions. We appreciate your insights and address the issues you have highlighted. The pages and line numbers mentioned by the authors below refer to the ones in the revised manuscript without track changes. Besides the reviewers' comments, we have also made slight changes in the revised manuscript for consistency and wording. At the same time, the title of the manuscript was slightly modified to make it simpler from "The potential to convert methane to organic acids is a widely found trait among gammaproteobacterial methanotrophs of freshwater lake and pond ecosystems" into "Conversion of methane to organic acids is a widely found trait among gammaproteobacterial methanotrophs of freshwater lake and pond ecosystems"

Reviewer #1 (Comments for the Author):

This manuscript describes a useful and generally well presented study that shows organic acid production in two isolates of methanotrophs representing genera in which this activity had not previously been seen. There is also useful discussion of genomic and metagenomic information relating to this activity in the isolates and beyond. A number of issues require attention as indicated below.

1. Line 105. Since two strains are referred to here, the apostrophe should come after the final s at the end of this line.

Response: Thank you for spotting the error. We have modified it accordingly (See page 5, line 104)

2. Lines 134 to 139 and Figure 1. I could not find evidence in the figure to support the statements here about the different effect of replenishing the methane/air gas mixture on the two strains characterised in this work. The figure shows only one set of data for optical density for each strain, so the effect of omitting the gas replenishment cannot be judged. Also, the simple statement in the text that replenishment increases organic acid production in strain S1L but not in strain S2AM does not adequately summarise what the data show. In fact, in strain S1L gas replenishment results in more succinate but less malate compared to what happens without replenishment. In strain S2AM it is true that replenishment generally does not increase organic acid production, but there is a sharp spike in lactate concentration at 20 h when no replenishment is done. The OD data with and without has replenishment need to be shown and more accurate description of the data is needed. If the lactate spike in strain S2AM is an artifact (rather than a reproducible phenomenon), some additional experimental work would be needed. Statistical tests to validate any important changes commented upon would be helpful.

Response: Thank you for bringing up the important points. We have revised Figure 1 to show the optical density (OD) change in both treatments. As seen, the OD is higher for S1L in treatment where gases were replenished. But, as reported, this was not the case with S2AM.

Furthermore, we now also provide a new figure in the Supplementary data file (See Figure S4 below), to show the change in the total sum of organic acid-carbon in time, which further shows that the gas replenishment increased organic acid production of S1L but not that of S2AM (See new Figure S4 below). As noted by the referee, it is true that there are also qualitative changes, i.e., different types of organic acids change differently. But, considering our main aim, we have decided not to discuss those in more detail. The main aim of this study is to show that the strains produce organic acids and

also to provide information on what kind of organic acids they produce. For the lactate production of S2AM, the most important result is that we confirm that it produces lactate as shown in experiments presented in Figure 1 but also in the T and pH gradient tests shown in Supplementary Fig. S3.

We have also added a two-sample T-test comparing Optical density and total organic acid-carbon concentration in the liquid medium between conditions with and without CH₄ and air replenishment of both strains S1L and S2AM in the file of Supplementary data for Figure 1 and Figure S4. The results of t-test are also summarized in manuscript on page 6, lines 136-143.

Figure S4 is mentioned in the main manuscript on page 6, lines 138 and 141.

Figure S4. Total organic acid-carbon concentration in the liquid medium during 20- and 34-day incubation for A) *Methylovulum psychrotolerans* S1L and B) *Methylomonas paludis* S2AM, respectively, in both tests with and without CH₄ and air replenishment.

3. Legend of Figure 1. The red line referred to in parts E and F is horizontal, not vertical.

Response: Thank you for pointing out an error. We have modified it as suggested. Please refer to the revised caption of Figure 1 for details.

4. Line 145. To cover the possibility that lactate dehydrogenases might exist that cannot currently be identified by bioinformatics, I suggest inserting "an identifiable" before "lactate".

Response: Thank you for your suggestion. We have added "an identifiable" (page 7 line 149). Furthermore, to "soften" the next sentence, we also reworded "Instead, we propose that lactate excretion is likely from methylglyoxal/2-oxopropanal detoxification generally occurring in microorganisms" into "It is possible that lactate excretion is from methylglyoxal/2-oxopropanal detoxification generally occurring in microorganisms (17, 18)" (See page 7, lines 150-151)

Additionally, we have tried out a genome annotation analysis using METABOLIC (Zhou et al., 2022) (not mentioned in manuscript); however, it did not either find any identifiable *ldh* in the genome of S2AM.

Reference:

Zhou, Z., Tran, P.Q., Breister, A.M., Liu, Y., Kieft, K., Cowley, E.S., Karaoz, U., Anantharaman, K., 2022. METABOLIC: high-throughput profiling of microbial genomes for functional traits, metabolism, biogeochemistry, and community-scale functional networks. *Microbiome* 10, 33. <https://doi.org/10.1186/s40168-021-01213-8>

5. Line 148. Insert "the products of the" between "by" and "gloA".

Response: Thank you. We have added it accordingly as suggested. See page 7, line 152

6. It would be helpful to include, maybe in the supplementary material, a figure showing the pathways from methane to organic acids together annotated to show which genes have been found in the genomes of strains S1L and S2AM.

Response: Thank you for your suggestion. We agree with your suggestion and create the possible pathway of CH₄ oxidation into organic acids during O₂-limiting conditions. We have added the Figure S5 as below to Supplementary data file.

We have mentioned Figure S5 to the main manuscript (page 6, line 145-146) as follows:

“The genes encoding putative enzymes driving the organic acid production were found in the genomes of both strains (Supplementary Table S1, Fig. S5).”

Figure S5. Proposed pathway for organic acid production during methane oxidation under A) oxic and B) hypoxic conditions in aerobic gammaproteobacterial methanotrophs (gMOB), i.e., *Methylovulum psychrotolerans* S1L and *Methylomonas paludis* S2AM. The genes annotated for involvement in organic acid production, as found in the genomes of both S1L and S2AM strains, are listed in Supplementary Table S1.

7. Mindful that methanotrophs have been considered for conversion of methane into multicarbon

acids and other molecules, it would be interesting to calculate the maximum carbon atom percentage of methane to organic acid achieved in this work.

Response: Thank you for your suggestion. We agree that it is good to add carbon atom conversion from CH₄-C into organic acids. So, we have added the carbon conversion efficiency of consumed methane into various organic acids as well as total organic-acid carbon in Table 1.

8. Table S1. The information provided about the gene functions from the isolated strains is very clear, but that from the MAGs are given as KO numbers without gene names or putative functions. The table would be more easily accessible if some information about gene name or putative function were added for these.

Response: Thank you for your suggestion. We agree that providing gene names and putative functions for the MAGs would be beneficial. Accordingly, we have identified the KO numbers and their putative functions and added them to Supplementary Table S1, similar to what we have done for the isolated strain.

Reviewer #2 (Comments for the Author):

In this work, the authors investigate the fermentative abilities of two isolated strains of methanotrophic gammaproteobacteria, *Methylomonas paludis* S2AM and *Methylovulum psychrotolerans* S1L. This work follows up on a recent publication from the same authors (10.1038/s43705-022-00172-x) that reports on the fermentative ability of a *Methylobacter* isolate from the same boreal lake environment. These studies are important because they provide molecular information about the ability of methanotrophs to support non-methanotrophic organisms in the environment via methane gas. The work uses the largely the same methods as the previous publication, and the conclusions appear sound and are not overstated.

Comments:

Figure 1:

-I think it would be helpful to put the organic acid legends in panels I and J as well.

Response: Thank you for your suggestion. We acknowledge its importance and have included the organic acid legends in panels I and J as suggested.

-A description of the error bars is not provided. Are these means and standard deviations? How many replicates? This is also true in figures S2 and S3.

Response: Thank you for your suggestion. We have provided descriptions of the error bars in the captions for Figures 1, S2, S3, and S4.

-In E and F, the Oxygen:Methane ratio appears to dip below 1 in some cases, which is unusual for aerobic methanotrophic metabolism. The authors should provide an explanation for this.

Response:

Thank you for bringing attention to this issue. The batch cultivation approach, involving the supply of CH₄ and air in the headspace, could lead to O₂ limitation over time. In these figures, we aim to show the overall trend in the consumed Oxygen:Methane ratio for these methanotrophs. As a result, we observed that the ratio dipped below the stoichiometric ratio for aerobic CH₄ oxidation but occasionally also indeed below 1, but generally it was around 1. We believe this could indicate the potential for organic acid production through either fermentation or microaerobic methane oxidation. We have elaborated on this point in the revised manuscript on page 6, lines 133-136 as follows:

“The average consumed O₂/CH₄ ratio (~1.0) was below the stoichiometric ratio in aerobic CH₄ oxidation (Fig. 1E,F). This indicates that O₂-limited CH₄ oxidation (during hypoxic conditions) initiated the accumulation of organic acids (4, 10).”

The fluctuation in the O₂/CH₄ ratio could also arise from the calculations, given that we observed low amounts of consumed CH₄ and O₂, ranging from 0.007 to 0.5 mmol, in this study. These low concentrations introduce additional variables that could affect the ratio. Nonetheless, the main point here was to show that the ratio was reduced below the stoichiometric ratio of aerobic CH₄ oxidation.

Methods:

-It would be nice to have more information about how the organic acid quantification was performed (mobile phase, gradient, concentrations of standards, etc.) since this is the paper's main point.

Response: Thank you for your suggestion. We agree that it is important to add more information of organic acid quantification. We have added text in the main text (page 5, line 114-115) as follows:

“(See detailed methods in Supplementary Information)”

Due to there is word limited in the main manuscript, We have added in the section ‘Analytical methods’ in the Supplementary Information as follows: “Specifically, the mobile phase consisted of 0.01 N H₂SO₄ with a flow rate of 0.4 mL min⁻¹ and column temperature was set at 70 °C. Standard solutions were prepared using sodium acetate, sodium formate, sodium succinate, sodium lactate, and sodium malate with concentrations ranging from 0 to 5 mM.”

-I suggest putting the description from the top of table S1 in the methods section for how genomes and MAGs were searched for fermentative genes.

Response: Thank you for your suggestion. We modified the description to fit in the materials and methods section as follows:

page 5, line 108-109: “...[MAGs assembled and taxonomically annotated by Buck et al. (14)]...”

page 5, lines 112-113: “We were specifically focused on the key genes encoding enzymes involved in organic acid and H₂ production.”

September 9, 2023

Dr. Antti Juhani Rissanen
Tampereen yliopisto - Hervannan kampus
Faculty of Engineering and Natural Sciences
Korkeakoulunkatu 6
Tampere FI-33720
Finland

Re: Spectrum01742-23R1 (Conversion of methane to organic acids is a widely found trait among gammaproteobacterial methanotrophs of freshwater lake and pond ecosystems)

Dear Dr. Antti Juhani Rissanen:

Your manuscript has been accepted, and I am forwarding it to the ASM Journals Department for publication. You will be notified when your proofs are ready to be viewed.

Sincerely,

Jannell Bazurto
Editor, Microbiology Spectrum
